# The Controversial History of Hormone Replacement Therapy

**DOI:** 10.3390/medicina55090602

**Published:** 2019-09-18

**Authors:** Angelo Cagnacci, Martina Venier

**Affiliations:** Obstetrics and Gynecology Unit, Azienda Sanitaria Universitaria Integrata di, 33100 Udine, Italy; martina04venier@gmail.com

**Keywords:** hormone replacement therapy, osteoporosis, breast cancer, cardiovascular disease

## Abstract

The history of hormone replacement therapy (HRT) started in the 1960s, with very high popularity in the 1990s. The first clinical trials on HRT and chronic postmenopausal conditions were started in the USA in the late 1990s. After the announcement of the first results of the Women’s Health Initiative (WHI) in 2002, which showed that HRT had more detrimental than beneficial effects, HRT use dropped. The negative results of the study received wide publicity, creating panic among some users and new guidance for doctors on prescribing HRT. The clear message from the media was that HRT had more risks than benefits for all women. In the following years, a reanalysis of the WHI trial was performed, and new studies showed that the use of HRT in younger women or in early postmenopausal women had a beneficial effect on the cardiovascular system, reducing coronary disease and all-cause mortality. Notwithstanding this, the public opinion on HRT has not changed yet, leading to important negative consequences for women’s health and quality of life.

## 1. From the Past to the Present

In the 20th century, the interest in menopausal disorder has been increasing, but it took a long time to realize the real impact of hormone depletion on women’s health. The clinical conditions associated with menopause were identified as “Hormone Deficiency Syndrome” [1], which included, besides hot flashes, other late onset chronic diseases such as osteoporosis, cardiovascular events, Alzheimer’s disease, and vaginal atrophy. At the beginning of the 20th century, the Food and Drug Administration (FDA )approved an estrogen product for the first time, Preamirin^®^, for the treatment of hot flashes [2]. 

The feminist movement in the 1960s changed women’s status and life expectancy, encouraging menopausal therapy, especially in European countries, with the concept of “feminine forever”. Wilson’s book, published in 1966 (“feminine forever”), became a bestseller, with its claim that “menopause is a hormone deficiency disease, curable and totally preventable, just take estrogen” [3]. Hormone replacement therapy (HRT) was presented as a therapy that could allow women to free themselves from the malediction of estrogen loss and conserve their femininity. In the 1970s, the finding that unopposed estrogen supplements were associated with an increased risk of endometrial cancer had a bad impact on HRT’s reputation [4,5]. Ziel et al. observed a probable connection between the administration of conjugated estrogen alone and the development of endometrial cancer [4].

Nevertheless, in the following years, researchers discovered that reducing the dosage of estrogen and combining it with progesterone could reduce the risk of endometrial cancer [6]. Such combined therapy was recommended for women with an intact uterus, raising renewed enthusiasm for HRT treatment. 

The FDA initially approved HRT only for the treatment of hot flashes and not for the prevention of chronic conditions, but in 1988 the prevention of osteoporosis was included among the FDA-approved indications [7,8].

Furthermore, in the same years, numerous observational studies [9,10,11,12] suggested that HRT had various benefits, not only with regard to the treatment of menopausal symptoms, but also in the prevention of chronic diseases. Thus, the idea switched from “feminine forever” to “healthy forever”. The use of HRT increased further, and the American College of Physicians developed the first guidelines for using HRT as a preventative therapy for the chronic diseases of postmenopausal women [13,14], even if experts raised some concern about the impact that progestin may have on estrogen benefits, particularly in the cardiovascular system [15]. 

On the basis of the claimed positive effect of HRT on cardiovascular health, the FDA required that this supposed HRT-induced cardiovascular benefit be confirmed by randomized clinical trials.

The first randomized clinical trial was a study of secondary prevention of cardiovascular disease, the Heart and Estrogen/Progestin Replacement Study (HERS) [16]. This study enrolled 2736 postmenopausal women with established coronary heart disease. Enrolled women were randomized to receive HRT (0.625 mg of conjugated equine estrogens and 2.5 mg of medroxyprogesterone acetate) or a placebo. No difference was found between the two groups after four years of follow-up, but in the HRT group, an increase of coronary events (nonfatal myocardial infarction or coronary heart disease (CHD) death) was found after one year. This induced increase declined in the subsequent years [16].

In 1998, the Women’s Health Initiative (WHI) was started, which was the largest randomized study to date that was aimed at evaluating the effect of HRT on the most common causes of death and disability in postmenopausal women, such as cardiovascular disease, cancer, and osteoporosis. Women with uteri (16,608 participants) were randomized and received a combination of 0.625 mg Conjugated Equine Estrogens (CEE) and 2.5 mg of medroxyprogesterone acetate, and women without uteri (10,739 participants) were randomized and received 0.625 mg of conjugated equine estrogen or a placebo. The first results of the WHI were published in 2002 after a mean follow-up period of 5.2 years. In the group with intact uteri, an increased incidence of coronary heart disease and breast cancer was observed in concomitance with a reduction of osteoporotic fractures and colorectal cancer [17]. Given these results, it seemed that the risks outweighed the benefits, and the trial was prematurely discontinued. The data were largely disseminated to the media, creating panic among HRT users and forcing new guidance for doctors on prescribing HRT. The message was that HRT, with no specification of type and route of administration, was associated with more risks than benefits. However, no distinction was made between users and their age.

The trial with only estrogen (performed in hysterectomized women) continued, and the preliminary data were published in 2004 [18]. In addition, this trial was stopped prematurely after 6.8 years of follow-up due to evidence of a small increased risk of ischemic stroke in the absence of other significant cardiovascular benefits [18]. In spite of the benefits (such as a reduction of osteoporotic fracture and colon cancer) and of the not-increased risk of breast cancer or cardiovascular disease, the overall message on HRT remained negative. After these announcements, the UK regulatory authorities issued an urgent safety restriction about HRT, recommending that doctors should prescribe the lowest effective dose for symptom relief, should use it only as a second-line treatment for the prevention of osteoporosis, and should not use it in asymptomatic postmenopausal women.

Since that time, there have been ongoing discussions and controversies about the WHI design and conclusions, and many of its results have been extensively debated. It emerged that one important limitation of the WHI was that most of the participants were more than a decade past their final menstrual period, raising the question of whether the results of the trial could be applied to younger women. Furthermore, the WHI tested only CEE either alone or in combination with a single progestin, medroxyprogesterone acetate. As a result, the WHI findings did not answer doubts about the safety and effectiveness of other HRT formulations, regimens, and delivery methods. Despite its limitations, the WHI had a negative impact on the global perception of HRT, leading to a marked decline in HRT utilization. Many doctors stopped prescribing HRT, and many women abandoned HRT immediately.

It has been suggested that the conflicting findings between the WHI and previous observational studies were the consequence of the different ages of the enrolled women. Some observational studies had included symptomatic women who had started HRT near the onset of menopause, while women enrolled in the WHI trial were asymptomatic, older (average age 63.2), and frequently more than 10 years on from the start of menopause. It has been suggested that a “window of opportunity” may exist, a period of time close to menopause where the benefits of HRT exceed the risks [19]. 

The follow-up of the WHI continued for 13 years, and some results, including an age stratification of the cardiovascular outcomes, were published in the following years [20,21,22,23]. A reanalysis of the WHI trial with new studies and a metanalysis showed that the use of HRT in younger women (50–59 years) or in early postmenopausal women (within 10 years of menopausal onset) had a beneficial effect on the cardiovascular system, reducing coronary diseases and all-cause mortality [22,24,25,26,27]. Furthermore, a large controlled trial from Denmark (reported in 2012) demonstrated that healthy women taking combined HRT for 10 years immediately after menopause had a reduced risk of heart disease and death from heart disease [28]. Unfortunately, these data did not receive appropriate coverage by the media, and the fear regarding HRT has persisted.

Thus, the story of HRT over the years has shown a trend with two peaks of utilization: a first rise in the 1960s and a second, higher increase in the years 1999–2000, before the publication of the WHI data. After these years, there was a precipitous decline in HRT use in many countries [29]. For instance, the use of HRT dramatically declined by 46% in the USA [30] and by 28% in Canada [31], and similar data were observed in European countries such as Germany [32,33] or the United Kingdom [34]. 

If HRT was really harmful to the health of postmenopausal women, its massive reduction in use should have resulted in an improvement in women’s health. Observational studies have tried to show trends of hormone-related cancers and chronic conditions following the WHI publications. Cancer and chronic conditions are multifactorial, and modification of a single factor, such as HRT use, may not result in a clear epidemiological change. Still, the analysis of epidemiological trends or modifications in the post-WHI era may give additional insights into the role played by HRT in women’s health (Table 1). 

## 2. Breast Cancer

### 2.1. Effect of HRT

Over the years, data regarding the impact of HRT on breast safety and breast cancer mortality have been controversial. Most of the meta-analyses and observational studies performed in the 1990s reported no increase in the risk of breast cancer with estrogen use [35]. However, some increased risks related to dose and duration of use were found with the administration of combined estrogen–progesterone therapies [36].

The WHI reported an increased risk of breast cancer in the women treated with a combination of conjugated estrogen and medroxyprogesterone acetate [17], a risk that was significantly higher than in placebo users after 5.6 years of treatment (approximately 6–7 years) [37,38]. Vice versa, the risk of breast cancer was significantly lower than in placebo users for those women treated with only conjugated estrogen [37,39]. One important, but incorrect, thought reported by the media after the WHI trial publication was that HRT causes breast cancer “de novo”. However, this study did not distinguish between de novo tumor development and the growth of an occult tumor present before the beginning of therapy. After analyses of the biology of occult tumors, current opinion about breast safety with HRT has changed. The modern thinking is that estrogen could have a promotional and noncarcinogenic effect on occult tumor cells and that this effect is probably greater with an estrogen–progesterone therapy [40]. 

### 2.2. Consequence of HRT Withdrawal

Following the publication of the WHI results and the associated drop of HRT use, some envisioned a concomitant decline of breast cancer incidence. However, the impact of HRT cessation showed an important variability between nations. Clarke and colleagues were the first to publish data about the incidence of breast cancer between 2001 and 2003 [41]. The authors reported that after the WHI publication, the use of HRT in northern California declined by 68%, and at the same time, breast cancer incidence declined by 10%. A study performed in California between 2001 and 2004 [42] similarly hypothesized that a decline in breast cancer incidence was linked to reduced HRT use. These data were in accordance with those observed in the Surveillance, Epidemiology, and End Results (SEER) study, which showed a 6.7% decrease in breast cancer incidence in 2003 [29]. Other western countries documented a reduction in breast cancer incidence after the reduced use of HRT [43]. In Germany, breast cancer incidence declined by 8.8% annually from 2002 to 2005, with an almost 50% reduction in HRT use [44]. In France, the breast cancer rate decreased by 14.7% from 2003 to 2007, with a reduction in HRT use from 32% to 11% between the years 2001 and 2007 [45]. In Australia, a 6.7% decline in breast cancer incidence was observed in 2003, with a simultaneous 8% reduction in HRT use between 2001 and 2003 [46]. In Canada, an 8% decreased incidence in breast cancer was observed every year from 2002 to 2004, with a 15% reduction in HRT use [47]. 

In Italy, the Netherlands, and Spain, the absolute decline of therapy after 2002 did not translate into a decline in breast cancer rates [43]. In addition, in the UK, the use of HRT showed a peak of 25% in 2000 and decreased thereafter. In spite of that, breast cancer incidence declined 0.8% per year from 1999 to 2006 [48]. Interestingly, the incidence of estrogen receptor-positive breast cancer in the USA increased to a maximum in 1999, with a downward trend starting in 2000, at least two years prior to the WHI publications [43,49]. Moreover, after some years, there was a new rise in breast cancer incidence (for both lobular and ductal breast cancer), such that by 2012, the rates were like those of 2001. This could mean that other factors besides HRT can explain the trend in breast cancer. Likely, some of these data can be explained by changes in the national screening programs for breast cancer. High rates of mammography, which has been extended to younger women, may have resulted in a greater incidence of breast cancer in the years following screening implementation, followed by a decline of breast cancer incidence as a consequence of the early diagnoses of the previous years [50,51].

## 3. Endometrial Cancer

### 3.1. Effect of HRT

The use of unopposed estrogen has been associated with an increased risk of endometrial cancer [4]. Consequently, guidelines have indicated that in women with uteri, estrogen therapy should be prescribed together with a progestin molecule [8]. Evidence of reduced endometrial cancer risk with combined HRT was well represented in the HERS trial, which showed seven fewer endometrial cancer cases per 10,000 in women using HRT than in placebo users [52]. Similarly, the Million Women Study reported that combined HRT therapy reduced the risk of endometrial cancer [53]. Continuous combined therapy has better endometrial safety than sequential therapy does [54,55]. A large Finnish case–control study [56] showed that sequential therapy induced an elevated risk of endometrial cancer when used for longer than 10 years. However, the roles of different types of progestagen remain to be assessed. In two observational studies, the most commonly used progestin molecules, norethisterone acetate and medroxyprogesterone acetate, did not differ in their protection from endometrial cancer [53,56]. On the basis of the literature, oral micronized progesterone administered sequentially for 12–14 days/month at a dose of 200 mg/day is effective for endometrial protection for up to five years of use [57]. However, for treatments longer than five years in the E3N cohort study [58], combined therapy with oral micronized progesterone or dydrogesterone was associated with an increased risk of endometrial cancer [59].

### 3.2. Consequences of HRT Withdrawal

After the WHI publication and the drop in HRT prescription, endometrial carcinoma rates increased between 2001 and 2012 [60]. Similarly, in England, Martin and colleagues observed a change in endometrial cancer mortality after 2002, with nine additional endometrial cancer deaths per year [51]. 

## 4. Cardiovascular Disease

### 4.1. Effect of HRT

The incidence of cardiovascular events increases in postmenopausal women, particularly in those experiencing severe vasomotor symptoms. The Study of Women’s Health Across the Nation (SWAN) showed that women with hot flashes had higher subclinical CVD, including greater aortic calcification, poorer endothelial function, and higher intima media thickness, than did women without menopausal symptoms [61,62,63]. Clinical studies have shown that HRT favorably impacts the risk factors for CVD [64,65,66,67,68,69]. The clinical efficacy could be related to early treatment, but also could be related to the administration of HRT to those with a higher cardiovascular risk due to their symptoms. This was not the case in the WHI, where HRT was given years after menopause and to asymptomatic women. Thus, it is not surprising that the results of the WHI trial did not confirm a protective effect of HRT against coronary heart disease, stroke, and venous thromboembolism [17]. However, a reanalysis of the data using age stratification reported that in women within 10 years of the onset of menopause, the administration of HRT decreased the rate of coronary artery disease and all-cause mortality [22]. Two subsequent meta-analyses, which used cumulated data from 23 and 30 randomized clinical trials, respectively, reported a decrease in CVD and all-cause mortality in HRT users younger than 60 years of age or in those who began menopause less than 10 years prior [24,25]. More recent randomized trials have tested the timing hypothesis, the Early versus Late Intervention with Estradiol (ELITE) study [70] and the Kronos Early Estrogen Prevention Study (KEEPS) [71], and have reported a protective effect of HRT in young women close to menopause. More recently, the Danish Osteoporosis Prevention Study (DOPS) reported the same findings after 10 years of treatment with HRT versus a placebo [28].

### 4.2. Consequences of HRT Withdrawal

From the above data, it could be expected that a lack of treatment of symptomatic postmenopausal women may have translated into an increase in cardiovascular events after 2002. A Finnish study reported increased deaths from myocardial infarctions and strokes in women discontinuing HRT, especially in those younger than 60 years of age [72]. In the USA, a report comparing male and female mortality between two periods, the mid-1990s and between 2002 and 2006 [73], reported a reduction in male mortality but an increase in female mortality in the same timeframe. An English ecological study reported unfavorable changes in myocardial infarctions and strokes subsequent to the HRT fall following the WHI publication [51]. In contrast, one ecological study performed in the USA after 2002 reported a correlation between the fall in HRT use and a reduced incidence of acute myocardial infarction [74].

## 5. Fractures

### 5.1. Effect of HRT

Postmenopausal osteoporosis is a consequence of estrogen withdrawal in which inflammation plays a multifactorial role [75,76]. Clinical studies have unanimously shown that any kind of HRT is capable of reducing bone turnover, reducing bone reabsorption, and increasing bone mineral density. Both high bone turnover and low bone mineral density are risk factors for bone fractures. The supposed preventive effect of HRT on bone fractures has been reported by many observational studies [77,78]. The WHI was the first randomized trial that showed the clear efficacy of HRT in reducing vertebral and hip fractures by about 34% [17]. The same data were obtained in the DOPS study [79]. The effect was demonstrated in a population of normal women that was not specifically suffering from osteoporosis, i.e., a clinical situation where other antiosteoporotic therapies failed to show a preventive effect.

### 5.2. Consequences of HRT Withdrawal

The follow-up to the WHI study showed no increase in fractures five years after HRT withdrawal [80]. In another observational study (after 15 years of follow-up), conserved beneficial effects on bone were reported in women who used HRT for more than five years [81]. However, in a large study, Islam and colleagues found that the risk of fractures in postmenopausal women increased in the three years following the publication of the WHI [82]. An Italian analysis estimated 43,000 extra bone fractures per year in the USA that were associated with decreased HRT use [83]. A more recent longitudinal observational study concluded that women who discontinued HRT had a higher risk of fractures compared to those who continued therapy, which has an important effect on women’s health [84]. 

## 6. Conclusions

The controversial history of HRT is about the history of a powerful pathogenetic therapy for all postmenopausal disturbances. Its effects on symptoms are and were immediately visible, at first prompting rapidly growing estrogen use. A lack of knowledge about its side effects and complications, particularly in the endometrium, prompted consequences that limited HRT use. Subsequent association with progestin allowed for the widespread use of HRT, with favorable consequences on many aspects of women’s health. Unfortunately, the surge in HRT use and its consolidation was abruptly stopped by the publication of the WHI trial, which was inadequately designed, evaluated, and reported. The damage done was huge, basically leaving many symptomatic women without an effective treatment, even if the epidemiological data were not strong enough to document a clear harm to women’s health. Although most of the evidence obtained was only with oral conjugated estrogen with or without medroxyprogesterone acetate, further studies and analyses have consolidated the view that HRT is highly beneficial when given to symptomatic women within 10 years since the onset of menopause or to symptomatic women that are under 60 years of age. However, the damage remains, and low HRT use, which is unjustified, continues to occur throughout the world. 

## Figures and Tables

**Table 1 medicina-55-00602-t001:** Epidemiological modification in the incidence of hormone-related cancers, cardiovascular disease (CVD), and bone fractures in the post-Women’s Health Initiative (WHI) era.

Disease	Modification
Breast Cancer	**Decreased** or Unchanged
Endometrial Cancer	**Increased**
CVD	**Increased** or Unchanged
Bone Fractures	Increased or Unchanged

In bold is the stronger evidence.

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
