# Peer review of "The Controversial History of Hormone Replacement Therapy"

_medicina, 2019, doi:10.3390/medicina55090602_

Round 1

Reviewer 1 Report

Review article with the tittle “The Controversial History of Hormone Replacement Therapy, Hormone replacement therapy: Benefit and Risks” describes about the milestone of hormonal therapy in the state of hormonal depletion, especially in the treatment of estrogen therapy for menopause. Otherwise, it explains about the disease-related effects caused by hormone replacement therapy (HRT). This topic is quite old, but the problems-related to the topic are so complicated.

Novelty and significance

This article possesses less quality of being new and original. There have been similar topics to be addressed since 2003. Otherwise, there are certain hormonal therapy which are related to the sex hormones (male and female-related). And the authors just focus on female hormonal therapy. However, it is better if the authors add more references about the theoretical and clinical problems related with hormone replacement therapy, and even about the treatment of the disturbances caused by (HRT). Their article is too brief with only 2 subtitles; from the past to the present and consequences of HRT withdrawal. 

Specific comments

Authors without academic title.

Affiliation number 2 is unavailable.

All references published year less than 2017. Authors need to read new papers.  

Author Response

-The topic of the article was about postmenopausal hormonal therapy.

-The aim of the article was to address the main data leading to an increase or a decrease of HRT treatment.  The article was based on how HRT was perceived on the basis of real events. Theoretical data are not part of this article. The clinical benefits of HRT, i.e. those clearly defined such as improvement of hot flushes, prevention of osteoporosis and eventually of cardiovascular disease are mentioned. it is not the aim of the manuscript to address and discuss more controversial issues such as dementia, psychological disturbances or urogenital effects.

-The text was further divided in subchapters

- By further looking at the literature we did not find additional more recent references useful to support our narrative review

-Affiliation was corrected for the second author

Reviewer 2 Report

This paper is very nicely written. Numerous research studies were incorporated to summarise the history of HRT. It will be useful for other researchers to write their introduction of HRT related papers.

Author Response

We thank the reviewer for his comments.

Reviewer 3 Report

The authors briefly reviewed the history and controversies of HRT, especially after the publication of WHI study leading to a dramatic decrease of HRT use. The author subsequently correlated the changes in the occurrence of diseases following the drop in HRT use. It was assumed that “If HRT was really harmful to the health of postmenopausal women, its massive reduction in use should have resulted in an improvement in women’s health”. However, the studies of disease incidence cited are coincidental at best since the development of diseases are multifactorial. It is very hard to attribute to HRT use alone. Thus, the evidence presented might not be strong enough to make a valid conclusion that the discontinuation of HRT causes damage to women’s health.  

Author Response

We partially agree with the reviewer comments and we added his comments in the manuscript (please see lines 116-121).

Reviewer 4 Report

The study presents controversial history of HRT for some disturbances which has changeD over the years . The authors focused on especially two eras:  one  before HERS and WHI studies and the second after that trails .  The study is well written and describes the view of doctors  media and patients which changed after data presented in aforementioned studies. Authors focused especially on cardiovascular system,   endometrial and breast cancer . They wonder how  discontinuationg of HRT influenced the frequency of new incidents CVD and these two cancers. In that mansuscript the influence of HRT and stopping of the treatment   is not so widely discussed in relation to the osteoporosis.  There is also no data on ovarian cancer,  pulmonary cancer, colorectal cancer and Alzheimer's  disease  but probably that was not main aim of that manuscript.

Finally we can agree with authors ( supported by trials)  that HRT should be still given but with exact indications ( e.g menopausal symptoms) before 10 years of menopause and in women younger than 60 years old. What is important and what authors mentioned,  our view have source mainly from trials with CEE+ medroxyprogesteron acetate  what of course is not all  gamma of medicines in HRT especially in Europe. 

Author Response

The paper was mainly focused on facts that changed the perception and use of HRT. Accordingly, factors that did not impact on perceptions (such as ovarian cancer, pulmonary cancer etc. were not mentioned).

In the conclusion we mentioned that indications given by the reviewer for HRT use and further specified the limitation of the evidence to conjugated estrogens and progestins (lines 253-256)

Round 2

Reviewer 1 Report

Authors have corrected. 

Author Response

We thank the reviewer. We revised english language and spelling

Reviewer 3 Report

I am agreeable the modifications made by the authors, by cautioning the readers the changes in the incidence of the diseases discussed could be multifactorial, and that the cessation of HRT alone may not be enough to explain these changes. I request the authors to rephrase their sentences from 118-120 to make the meaning clearer to the readers. 

The manuscript will also benefit from a summary table detailing the incidence of diseases before and after the WHI report. 

Author Response

-The two sentences were rephrased. We hope that improved their clarity.

-A table was added.